# Beyond the Interface: Improved Pulmonary Surfactant-Assisted Drug Delivery through Surface-Associated Structures

**DOI:** 10.3390/pharmaceutics15010256

**Published:** 2023-01-11

**Authors:** Cristina García-Mouton, Mercedes Echaide, Luis A. Serrano, Guillermo Orellana, Fabrizio Salomone, Francesca Ricci, Barbara Pioselli, Davide Amidani, Antonio Cruz, Jesús Pérez-Gil

**Affiliations:** 1Department of Biochemistry, Faculty of Biology, and Research Institute “Hospital 12 de Octubre (imas12)”, Complutense University, 28040 Madrid, Spain; 2Department of Organic Chemistry, Faculty of Chemistry, Complutense University, 28040 Madrid, Spain; 3R&D Department, Chiesi Farmaceutici, 43122 Parma, Italy

**Keywords:** pulmonary surfactant, air–liquid interface, surface balance, surface-associated reservoir, interfacial spreading, interfacial film, pulmonary drug delivery

## Abstract

Pulmonary surfactant (PS) has been proposed as an efficient drug delivery vehicle for inhaled therapies. Its ability to adsorb and spread interfacially and transport different drugs associated with it has been studied mainly by different surface balance designs, typically interconnecting various compartments by interfacial paper bridges, mimicking in vitro the respiratory air–liquid interface. It has been demonstrated that only a monomolecular surface layer of PS/drug is able to cross this bridge. However, surfactant films are typically organized as multi-layered structures associated with the interface. The aim of this work was to explore the contribution of surface-associated structures to the spreading of PS and the transport of drugs. We have designed a novel vehiculization balance in which donor and recipient compartments are connected by a whole three-dimensional layer of liquid and not only by an interfacial bridge. By combining different surfactant formulations and liposomes with a fluorescent lipid dye and a model hydrophobic drug, budesonide (BUD), we observed that the use of the bridge significantly reduced the transfer of lipids and drug through the air–liquid interface in comparison to what can be spread through a fully open interfacial liquid layer. We conclude that three-dimensional structures connected to the surfactant interfacial film can provide an important additional contribution to interfacial delivery, as they are able to transport significant amounts of lipids and drugs during surfactant spreading.

## 1. Introduction

The pulmonary surfactant (PS) system is essential to avoid alveolar collapse in mammalian lungs, thus enabling the process of breathing. It is a surface-active material composed of a major fraction of lipids and a small fraction of four specific proteins: the hydrophobic SP-B and SP-C, essential for the biophysical function of surfactant at the interface, and the two hydrophilic proteins SP-A and SP-D, involved in lung immune defence. Once secreted by type II pneumocytes to the hydration layer coating the respiratory surface, PS rapidly adsorbs into the alveolar air–liquid interface and reduces the surface tension from ~70 mN/m (at 37 °C) to ~20 mN/m, stabilizing the alveolar surface [1]. The adsorbed surfactant film is formed not only by an interfacial monolayer but by a monolayer with multi-layered structures associated with it [2,3], thanks to the action of SP-B and SP-C [4]. These proteins also play a crucial role during breathing, facilitating the exclusion of less compressible phospholipids (PLs) and the formation of three-dimensional reservoirs upon compression (expiration), as well as promoting the re-spreading of the film during expansion (at inspiration) [5]. For more detailed information about surfactant composition, structure, and function, the readers are referred to some reviews [1,6,7,8,9]. Due to surfactant’s composition and the mentioned biophysical properties, it has been proposed that surfactant may serve as a powerful drug delivery vehicle in order to improve lung therapies.

The use of PS for different drug delivery strategies has been explored by many researchers with different targets [10,11,12,13]. Apart from the suitability of PS composition to efficiently solubilise poorly water-soluble molecules [14], the main reason why this material improves the distribution of drugs over the respiratory surface has to do with its capability to rapidly spread along air–liquid interfaces. Once adsorbed into the interface, surfactant spreads interfacially due to surface tension gradients, from regions where the surface tension is low (high concentrations of PS) to regions with higher surface tension, following what are known as Marangoni flows [15,16,17,18]. When therapeutic molecules are combined with PS, this lipid mass transfer implies the transport of the molecules associated with it; this has been called “the interfacial delivery” [12]. This process has been recently studied by our group [12,19,20] and other researchers [21,22] by using various designs based on surface balances. To the best of our knowledge, most of the designs were based on the one proposed by Yu and Possmayer [23], where a donor surface balance is connected with an acceptor one by a hydrated paper bridge that serves as an interfacial link. It has been demonstrated that only a surfactant monolayer is transferred from the donor to the recipient trough [24], meaning that the surface-associated reservoirs are not able to cross the paper bridge. As a consequence, the contribution of these structures to the transfer of lipids and thus, the delivered molecules, has been mostly disregarded.

To evaluate the potential transfer of surface-associated structures together with the spreading of the surfactant film over the interface, we have designed a new vehiculization surface balance where the interfacial paper bridge can be avoided. Two surface troughs, emulating upper and distal airways, are connected through a long and thin path that mimics the conductive airways and allows removing the paper bridge. We hypothesized that the amount of lipids and therapeutic molecules that are delivered over the air–liquid interface were underestimated in previous studies due to the bridge limitation. We therefore wanted to investigate the surfactant-mediated transport of drugs, not only by the interfacial layer, but also by the associated three-dimensional reservoirs. To study this, we used a hydrophobic drug, budesonide (BUD), as a model. This drug is a glucocorticoid with a potent local anti-inflammatory action typically used to treat or reduce the incidence of different lung diseases, especially in preterm infants [25,26]. The interaction of BUD with PS and the PS-promoted improvement in its distribution have been already confirmed by other researchers [10,20,25].

The main aim of this work was to study in more detail the spreading and interfacial delivery of drugs mediated by PS. To address it, we focused on (1) the contribution of surface-associated structures to the interfacial drug delivery, (2) how different materials travel along the interface and transport BUD with it, and (3) the spreading of PS reservoirs along an interface occupied by a pre-existing surfactant monolayer.

## 2. Materials and Methods

### 2.1. Lipids

Synthetic lipids dipalmitoylphosphatidylcholine (DPPC) and palmitoyloleoylphosphatidylglicerol (POPG), and the fluorescent lipid dye NBD-PC (phosphatidylcoline labelled with the fluorescent probe nitrobenzoxadiazole) were purchased from Avanti Polar Lipids, Inc. (Alabaster, AL, USA).

### 2.2. Budesonide

The corticosteroid budesonide was provided by Chiesi Farmaceutici S.p.A. (Parma, Italy). The synthesis, NMR, and MS spectra (Appendix A) of the red-fluorescent derivative of budesonide (F-BUD Appendix A; λ_ex_ = 590 nm; λ_em_ = 650 nm in methanol); namely, budesonide (BUD) in the whole work, labelled with an analogue of the fluorescent Nile Blue dye; are fully described in the Appendix A.

### 2.3. Pulmonary Surfactant

Bronchoalveolar lavages of freshly slaughtered porcine lungs were performed to obtain native pulmonary surfactant (NS). The material was purified from lavages as previously described [27]. The clinical surfactant *poractant alfa*, commercially available as Curosurf^®^, was obtained from Chiesi Farmaceutici S.p.A. (Parma, Italy). Organic extracts (OE) from NS or Curosurf were obtained by following the method described by Bligh and Dyer [28]. Essentially, the materials were subjected to subsequent washes with chloroform/methanol (2:1 *v*/*v*) and short centrifugations in order to collect the maximum amount of surfactant hydrophobic components at the bottom organic phase. The concentration of phospholipids in each material was determined by phosphorus mineralization, according to the protocol established by Rouser et al. [29]. The natural mixture of surfactant proteins SP-B and SP-C was purified from an organic extract of minced porcine lungs by a size-molecular exclusion chromatography in a Sephadex LH-20 matrix (GE Healthcare; Little Chalfont, UK) that separates the hydrophobic surfactant proteins from the lipids, as previously described [30]. Multilamellar vesicle (MLV) suspensions were prepared from lipid or lipid/protein mixtures or the full organic extract of surfactant by the evaporation–rehydration method. Proper volumes of the materials in chloroform/methanol (2:1 *v*/*v*) solution were dried under a nitrogen stream and under vacuum during 2 h to remove solvent traces. The dried films were reconstituted by hydration with a buffered solution (Tris 5 mM, NaCl 150 mM, pH 7.4) for 1 h, with vigorous shaking every 10 min. Large unilamellar vesicles (LUVs) of DPPC:POPG (7:3 *w*/*w*), with or without 2% *w*/*w* SP-B/SP-C, were formed by extrusion of MLV suspensions through polycarbonate membranes of 100 nm pore size (Whatman^®^ Nuclepore Track-Etched Membranes) with a mini-extruder (Avanti Polar Lipids Inc.). The incubation and extrusion temperatures were always set over the melting temperature (T_m_) of the lipids in the mixtures. To incorporate the fluorescent lipid NBD-PC or the corticosteroid into surfactant preparations, proper volumes of NBD-PC (1% mol/mol with respect to the total PLs) or BUD (1% or 2% *w*/*w* with respect to the total PL mass) diluted in chloroform/methanol were dried under a nitrogen stream first and then under vacuum to remove solvent traces. Then, aqueous surfactant suspensions (NS or Curosurf) were added to the dried film and incubated for 1 h at 37 °C. Organic extracts or lipids dissolved in chloroform/methanol were mixed and dried together with the fluorescent molecules to subsequently incubate with a buffered solution for 1 h at 45 °C.

### 2.4. Vehiculization Surface Balance

In order to evaluate whether membrane structures associated with the interfacial surfactant film could travel along the interface and transport drugs, we designed and built a new double-balance setup based on the one previously described by Yu and Possmayer [23] and that proposed by Hidalgo et al. [20], but avoiding the need to incorporate a paper bridge as a connection between donor and recipient compartments. This design consists of a donor trough (24 cm^2^) connected through a long path (90 cm, 75 cm^2^) to an acceptor trough (45 cm^2^), all engraved on a stainless-steel plate, emulating in a simplified way the continuity between upper, conductive, and distal airways, respectively. The depth of the path is shallower (3 mm) than the two troughs (5 mm) to minimize the diffusion of material from the donor to the receptor through the subphase. The two compartments and the path can be isolated one from each other by Teflon pieces designed to be introduced into two gaps placed at the entrance of both troughs. A surface pressure sensor is placed in each compartment in order to monitor changes in surface pressure occurring over time (accuracy of 0.1 mN/m). A schematic representation of this design is presented in Figure 1. The subphase consisted of 50 mL of a buffered solution (Tris 5 mM, NaCl 150 mM, pH 7.4) constantly thermostated at 25 ± 1 °C. In the experiments performed to compare the presence of the paper bridge, the recipient compartment was isolated from the rest of the system by one of the Teflon pieces and connected to the path by a bridge (1 × 5 cm) made of a hydrated filter paper (No. 1 Whatman filter paper), as also illustrated in Figure 1. Fifteen µL of the material to be assayed, at 20 or 50 mg/mL, were injected dropwise at the air–liquid interface of the donor compartment. When required, the air–liquid interface was pre-coated with ~10 µL of OE in organic solvent (1 mg/mL) deposited at the interface of the donor trough, in order to coat the interface of both compartments until reaching the desired surface pressure, but always <40 mN/m to avoid formation of surface-associated structures. After 10 min to allow for the evaporation of the organic solvent, 15 µL of the corresponding sample were injected at the donor subphase. Surface pressure changes in both troughs were recorded for 30 min. At the end of each experiment, the recipient trough was isolated from the rest of the system by the Teflon piece or by removing the paper bridge, as appropriate, and the recipient interface and 10 mL of the subphase were collected by aspiration. In this way, the interface-assisted surfactant vehiculization can be studied both in the absence or in the presence of the paper interfacial bridge. Experiments were performed in triplicate and data are represented as mean and standard deviation.

### 2.5. Fluorescence Spectroscopy

The spreading and transport capabilities of PS were evaluated by measuring the fluorescently labelled molecules reaching the recipient compartment. Fluorescence emission spectra (NBD-PC, λ_excitation_ = 466 nm, λ_emission_ = 532 nm; BUD, λ_excitation_ = 590 nm, λ_emission_ = 650 nm), were recorded at 25 °C in an Aminco Browman Series 2 spectrofluorometer. In order to determine the mass transfer of drug/lipids incorporated into different surfactant preparations from the donor to the acceptor compartment, values were interpolated into standard curves of BUD and NBD-PC at known concentrations.

### 2.6. Statistics

Data are expressed as mean and standard deviation. Result comparisons were conducted using one-way ANOVA, followed by Tukey’s post-hoc test or paired *t*-test as appropriate. *p* < 0.05 was considered to be significant. Analyses were carried out using GraphPad Prism 7 (v. 7, GraphPad Software, San Diego, CA, USA).

## 3. Results

### 3.1. Contribution of Surface-Associated Structures to the Interfacial Delivery

To determine whether the three-dimensional structures associated with the interfacial surfactant film are transported together with it during spreading, and therefore, could be implicated in the interfacial delivery of drugs, we used the novel design of a double-surface balance, as described in Section 2.4. To do so, we compared the spreading and transport of material from donor to the recipient compartment either by a direct connection (and no paper bridge) or upon connection of the path with the recipient compartment by an interfacial paper bridge. In the first scenario, we used aqueous suspensions of OE from NS doped with 1% mol/mol of a lipid dye, NBD-PC, at two different concentrations, 20 mg/mL and 50 mg/mL. As shown in Figure 2A,B, regardless of the presence of the paper bridge, OE suspensions are able to instantaneously adsorb at the donor air–liquid interface at the two concentrations tested and spread to the recipient compartment, producing a rapid increase in the surface pressure. In all cases, PS adsorbed to the interface and spread through long distances, producing surface pressure values near the equilibrium (40–45 mN/m) in both compartments, in just a few seconds. In these Π-time isotherms, we can also observe how in the absence of the paper bridge, the increase in surface pressure at the recipient compartment, occurring after the injection of the material into the donor trough, is almost immediate. However, in the presence of the bridge, there is a more conspicuous delay between the increase in pressure at the donor trough and that observed in the recipient trough (see Figure 2C). This delay occurs regardless of the surfactant concentration used, but it is slightly reduced when the concentration increases. Figure 2D represents the differences in final surface pressure between recipient and donor compartments in the different experiments. When the experiments were performed without the bridge, and all donor, path, and recipient compartments were interconnected by the subphase, the final surface pressures reached in both compartments were almost identical, or the recipient pressure was even higher. On the contrary, the presence of the paper bridge led to clear differences between donor and recipient pressures, probably indicating a limited transference of material (a sort of resistance to the interfacial current of material) into the recipient trough. However, no significant differences were observed in the final surface pressures reached at the recipient trough when comparing experiments in the absence or in the presence of the bridge (Figure 2E). Surfactant concentration does not seem to influence these parameters, as no significant differences were observed.

To confirm that the bridge was retaining the movement of some structures of the surfactant film and not only slowing down the spreading process, we collected the material at the recipient compartment to measure the fluorescence of NBD-PC. Thirty minutes after the injection of the sample, the recipient trough was isolated from the rest of the balance using a Teflon piece or by removing the bridge, and the interface plus 10 mL of the subphase were collected by aspiration. The amount of NBD-PC collected at the recipient interface was significantly reduced when the vehiculization experiments were performed in the presence of the paper bridge (Figure 2F). The recipient surface area accounts for 31% of the total area of the balance. This means that, if 3.15 µg and 7.9 µg of NBD-PC were injected at the donor trough in the experiments carried out with 20 mg/mL and 50 mg/mL PS, respectively, we could collect a maximum of 1 and 2.5 µg of NBD-PC in the recipient trough if all the material were adsorbed into the interface and we assumed a homogeneous distribution of the dye. However, a proportion of the material injected does not ultimately adsorb into the interface and is lost in the subphase. The presence of the bridge not only reduces the NBD-PC detected at the interface but also at the subphase of the recipient compartment (Figure 2G). When we collect the recipient interface by aspiration, we are probably taking some of the associated reservoirs together with the interfacial film, which increases the amount of material collected. However, we are likely also breaking some of the connections that maintain the 3D structures linked to the interfacial film, releasing them towards the subphase. Increasing PS concentration led to higher lipid transference, thus allowing higher NBD-PC collection both at the recipient interface and subphase, as revealed in Figure 2G,H.

As revealed in Figure 3, similar results have been observed using different surfactant preparations. In these experiments, we evaluated the effect of the paper bridge in retaining some surfactant structures upon spreading of various surfactant formulations. We tested whole NS, a purified porcine surfactant constituted by a complex mixture of multi-layered and multi-vesicular structures, and Curosurf, a clinical surfactant reconstituted from the hydrophobic fraction obtained from minced porcine lungs and possessing a simpler structure, similar to that of the OE previously assayed [31]. Figure 3A,B shows the adsorption/spreading isotherms derived from the injection at the donor trough of 15 µL NS or Curosurf (20 mg/mL), all doped with 1% mol/mol NBD-PC to follow the spreading of the material to the recipient trough in the absence (upper panels) or in the presence (bottom panels) of the paper bridge. When any of the surfactant preparations tested were deposited at the donor interface, an instant increase in the surface pressure to values of 35–45 mN/m was detected at the donor compartment. A few seconds or minutes later, depending on the presence of the bridge, a rapid increase in the recipient surface pressure to values near the value measured in the donor trough was also observed. As described before, the bridge caused a delay of ~2 min in the increase in the pressure at the recipient balance (Figure 3C). This delay was less than a minute in the case of the experiments carried out without the bridge. Figure 3D presents the difference in the final surface pressure between the recipient and donor troughs either in the absence or in the presence of the bridge. Larger differences were always observed when the bridge was used. However, as observed also in Figure 2E, no significant differences were detected in the final recipient pressure comparing the absence and the presence of the bridge (Figure 3E). The amount of NBD-PC collected from the recipient interface revealed a reduced lipid transference to this trough in the presence of the bridge for the three materials tested (Figure 3F). We detected higher fluorescence of NBD-PC at the recipient subphase of experiments performed without the bridge (Figure 3G), as observed before.

Finally, we wanted to disprove that the effects and differences observed could be due to a mere passive diffusion from the donor to the recipient trough along the volume of the whole subphase instead of being caused by an efficient interface-assisted spreading of the interfacial film and the structures associated. To do so, we performed similar experiments with purely lipidic LUVs (DPPC:POPG, 7:3 *w*/*w*) doped with 1% NBD-PC. The injection of 15 µL of the liposomes at 20 mg/mL into the donor compartment produced an insignificant increase in surface pressure, both with and without the paper bridge (Figure 4A). However, if we included 2% *w*/*w* (with respect to the total mass of PL) of the mixture SP-B/SP-C into the liposome composition, the material adsorbed to the air–liquid interface and spread rapidly along it to the recipient compartment, as observed in the Π-time isotherms in Figure 4B. At the end of each experiment, we collected the recipient interface and subphase. As a consequence of the poor interfacial adsorption of the pure lipid liposomes, we did not detect any NBD-PC fluorescence at the interface (Figure 4C) or in the subphase (data not shown) of the recipient compartment. In contrast, the incorporation of 2% of the hydrophobic surfactant proteins into the composition significantly improved the interfacial activity of the liposomes, substantially increasing the spreading to the recipient compartment, and therefore, the fluorescence detected at the recipient interface (Figure 4C). The introduction of the interfacial paper bridge produced the same effects in this material as the ones observed in the previous experiments, as shown in Figure 4B (bottom panel) and Figure 4C.

### 3.2. Transport of Budesonide over the Air–Liquid Interface by Different Materials

In a second scenario, we wanted to evaluate the vehiculization of the corticosteroid budesonide by different surfactant formulations. The low surface activity of BUD and its negligible detrimental effect on PS biophysical performance have been described by other authors [20,32,33,34]. In this case, we investigated the vehiculization of the drug not only through the interfacial film, as it was previously studied by Hidalgo et al. [20], but also by surface-associated reservoirs. A fluorescent derivative of BUD was combined with different PS samples at 1% or 2% *w*/*w* with respect to total PL, as described in the Section 2 for the different surfactant preparation used. Then, 15 µL of each of the sample at 20 mg/mL were deposited dropwise into the donor interface of the balance to monitor changes in surface pressure for 30 min. As revealed in the adsorption/spreading isotherms (Figure 5A), all PS preparations as well as proteo-lipid liposomes were able to efficiently adsorb into and spread over the air–liquid interface of the balance. The injection of the samples produced an instant increase in the surface pressure in both donor and recipient compartments. No deleterious effects on the surfactant biophysical properties were observed due to the incorporation of the corticosteroid in these experiments. As described in [20], PS can incorporate up to 10–20% BUD, depending on the content in cholesterol, without any significant impact on surfactant function, so our working percentage was far from causing any effect. Experiments were also performed in the presence of the paper bridge to connect the path with the recipient compartment, and the isotherms obtained were similar as the ones from previous experiments (Appendix A). After 30 min, we aspirated the recipient interface to collect the interfacial film and 10 mL extra of the subphase. Figure 5B shows the fluorescence of BUD detected at the recipient interface after each experiment. As occurred with the experiments performed with NBD-PC, BUD was better transported over the interface by organic extract suspensions (OE and OE from Curosurf). In the case of aqueous suspensions such as Curosurf, large amounts of BUD remained on the tube walls after incubation with it, indicating that the percentage of BUD incorporated into Curosurf membranes was well below 1% *w*/*w*. Considering this observation and the results obtained, we combined Curosurf and NS with 2% *w*/*w* BUD in order to maximize the amount of drug integrated into the sample. This increase in the drug content significantly amplified the signal of BUD detected at the recipient interface. No fluorescence was detected at the subphase of any of the experiments performed. Figure 5C represents the mass of BUD collected from the recipient interfaces. About ~1% of the total BUD initially added was recovered in the recipient trough in the experiments carried out without the bridge.

### 3.3. Spreading of PS over an Interface with a Pre-Existing Surfactant Monolayer

In an attempt to determine whether the surfactant reservoirs are able to insert into pre-existing interfacial films and spread along them, emulating what could occur in vivo when a potential exogenous PS would encounter the endogenous material, we performed experiments coating the interface with surfactant prior to the injection of the samples. To create a simple film coating the interface of the balance and avoid the formation of 3D structures, the OE in organic solvent was directly used to form the pre-existing film. Ten µL of OE (1 mg/mL) were applied dropwise onto the donor interface until the surface pressure in both compartments reached values of ~20 mN/m in order to coat the interface but not completely saturate it with surfactant. The organic solvent was allowed to evaporate during 10 min, and then 15 µL of the appropriate material were introduced into the donor subphase. After a further 30 min monitoring changes in surface pressure, the recipient interface and subphase were collected by aspiration. These experiments were carried out with fully interconnected compartments without using the paper bridge. Initially, we evaluated the vehiculization of BUD over an interface pre-coated with OE or Curosurf. As seen in Figure 6A,B, the injection of these two surfactant preparations produced an instant increase in the surface pressure of both compartments up to 30–40 mN/m, indicating that the injected material can adsorb and insert into the interface to complete saturation at equilibrium. Figure 6C shows the NileBlue-BUD fluorescence measured at the recipient interface. Even though there is a small fluorescence signal for the experiments with OE, this was only detected in one replicate out of three performed, and this is why the apparent standard deviation is high. No fluorescence was detected in any of the subphases. To determine whether increasing the surfactant concentration could affect the amount of fluorescence measured in the recipient trough, we performed experiments with OE doped with 1% NBD-PC at 20 mg/mL and 50 mg/mL. After coating the interface with a monolayer of OE, we injected 15 µL of OE + NBD-PC at the corresponding concentration at the donor subphase. Π-time isotherms show that OE at both concentrations was able to insert into the interfacial film and produce a surface pressure increase detected in both compartments (Figure 6D). No significant differences were observed in the surface pressure values obtained after the injection of OE at 20 mg/mL or 50 mg/mL. However, after the collection of the recipient interfaces and subphases from these experiments, we still could not detect any NBD-PC signal, even in the samples injected at 50 mg/mL.

## 4. Discussion

The combination of therapeutic agents with PS provides important improvements to target the respiratory system in inhaled therapies. The unique properties of PS make this material a powerful drug delivery system to efficiently incorporate poorly soluble molecules, protect them from degradation and clearance in the lungs, and ensure a proper distribution over the whole respiratory surface [35]. When the drug is combined with an exogenous surfactant, its potent spreading properties transport the drug together with the surfactant film along the air–liquid interface. As a result, drug concentration increases locally, reducing the doses needed, and likely the possible derived secondary effects [36]. For these reasons, several studies have proposed the use of PS as a delivery vehicle for different molecules, such as corticosteroids [25,37], antibiotics [38,39], or nanoparticles [13,40]. In this context, our group has studied the interfacial delivery of therapeutics mediated by PS, mainly using various setups combining different surface balances [12,19,20]. The surfactant spreading process has been studied and described by many researchers [41]; however, to the best of our knowledge, the role of surface-associated structures in the interfacial mass transfer has never been explored before. To do this, we designed a novel vehiculization surface balance and the results obtained showed that the three-dimensional structures associated with the interface spread along it, expanding the transference of surfactant and a surplus of therapeutic molecules over long distances.

When PS is secreted by the alveolar type II cells into the alveolar spaces, it rapidly adsorbs to the interface and forms the interfacial film including multi-layered structures associated with it, the so-called surface-associated reservoirs [2]. These structures contribute to stabilizing the film and act as a surplus of exchangeable material connected to the interfacial layer in order to ensure a properly functional film, especially under the elevated surface pressures reached during compression [8]. We wanted to investigate whether these reservoirs remained at the injection/adsorption site, from where they supply new material until saturating the whole interface, or whether they are also transported along the interface together with the interfacial film. In that case, they could also play an important role in the transport of drugs along the respiratory surface. To explore this idea, we performed experiments in our novel design of a vehiculizing surface balance comparing the absence and the presence of a paper bridge. In this work, the experiments have been performed applying surfactant at volumes and concentrations that ensured an excess of material to form surface-associated reservoirs. As demonstrated by Yu and Possmayer [24], just one monolayer of surfactant crosses the paper bridge and reaches the recipient compartment, even at high PS concentrations. After crossing the bridge, surfactant reservoirs in the recipient compartment are only formed upon film compression [12]. Differences in surface pressure between donor and recipient compartments when the bridge was present, as well as the delayed increase in the recipient surface pressure observed (Figure 2 and Figure 3), confirmed that the paper bridge, intended as an interfacial connector, imposes a clear resistance to surfactant spreading, which could include destabilization of the film if breaking the connections between the associated structures and the interfacial layer. In addition, the detection of lower amounts of lipid dye (NBD-PC) at the recipient compartment in the experiments with the bridge compared with those in its absence (Figure 2F–H), confirmed that the liquid-embedded paper somehow imposes a barrier to the travel of some surfactant structures. This lower signal was not due to any lipid selectivity in the interfacial film crossing the bridge, as previously demonstrated [23], and should be therefore due to a reduced transference of material as a whole. As the structure and density of the film depends on the type of surfactant and its concentration at the subphase [42], we examined whether higher PS concentrations could form more dense reservoirs, thus increasing the amount of lipids transported. Increasing OE concentration from 20 mg/mL to 50 mg/mL did not produce great differences in surface pressure values (Figure 2). Although no significant differences were observed, slightly higher fluorescence was detected at the recipient interface and subphase in experiments performed without the bridge when increasing OE concentration. Significantly higher amounts of the dye were collected at the recipient interface at 50 mg/mL (Figure 2F). Some differences were observed when comparing experiments with the bridge at both concentrations. The results of these experiments indicate that the strict interfacial monolayer exposed to air is probably very similar regardless of the concentration of surfactant used, leading to equivalent surface pressure values. However, the amount of material associated with the interfacial film increases at higher surfactant concentrations, leading to the transport of higher amounts of lipids over the interface. It may be possible that the thin hydration layer formed in the paper bridge is not enough to constitute a hypophase through which complex reservoirs could still travel. Alternatively, it could be possible that some interactions between surfactant complexes and the cellulose polymers in the paper contribute to retard and even prevent lateral diffusion of complex PS structures, acting as a kind of net that retains the material. Whatever the reason, it seems that if a complex film is formed at the donor compartment as a consequence of the adsorption of a high PS concentration, it does not raise the material reaching the recipient interface upon crossing the bridge (Figure 7).

The spreading-resistance effect imposed by the bridge over surfactant structures was observed in different surfactant preparations (Figure 3). In all formulations tested, the bridge produced a delay in surfactant spreading and reduced the amount of fluorescent probe reaching the recipient trough. Variations in the amount of NBD-PC detected upon delivery by the three materials tested were partly due to differences in the efficiency of dye incorporation into PS membranes, as it probably depends critically on the contact of the outer layers of the membrane structures in the aqueous suspensions with the drug films formed at the walls of the tubes, as previously described by Hidalgo et al. [20]. NS is an aqueous suspension with a multi-layered structure, which can hinder a proper and homogeneous incorporation of the dye into all its membranes. Curosurf is also an aqueous surfactant suspension, but its simpler structure could improve NBD-PC incorporation in comparison with NS. In the case of the OE, the dye was mixed with it in organic solvent solution, dried, and subsequently hydrated together, which facilitated an efficient and homogeneous incorporation of the fluorescent probe into surfactant membranes.

Liposomes of DPPC:POPG were used as a control without interfacial activity to verify that the material detected at the recipient trough was spreading necessarily across the interface and not passively diffusing through the subphase (Figure 4). These experiments showed that the possible contribution of a mere passive diffusion over the subphase should be considered as negligible, at least for the duration of these experiments. Our results demonstrated that the use of the paper bridge imposes a limitation for vehiculization experiments performed in surface balances and may induce an underestimation of the potential of PS spreading. Still, the resistance of the material of the paper bridge to PS lateral spreading could somehow mimic potential analogous effects that could be introduced by the mucus layer existing at the upper airways [43] and that should be explored. Further studies could be approached using this new balance and including mucins in the path to see whether they could also introduce a net resistance to lateral spreading of surfactant structures, a role that could be played by mucus in vivo when delivered exogenous surfactant travels from the trachea to the alveoli.

Surface-associated structures are also involved in the interfacial drug delivery process, as evidenced in Figure 5. Different surfactant formulations combined with BUD transported higher amounts of the drug in experiments carried out without the resistance played by the paper bridge. BUD was better vehiculized by OE formulations, as the incorporation method is more efficient in these cases, as discussed above. However, the results demonstrated that all surfactant formulations could transport the corticosteroid through the interfacial monolayer and by the surface-associated reservoirs. Despite the fact that the quantitative evaluation of the fluorescent probes transported entails some errors and limitations, mainly derived from the collection method, we always obtained a higher amount of NBD-PC (Figure 3) than of BUD reaching the recipient compartment (Figure 5). These differences denote that the lipid transference over the interface is more efficient than the transport of the drug, both by the interfacial film and the 3D structures. It will be of interest to determine, using this setup in a case-by-case basis, what is the relative intrinsic efficiency for different drugs to travel along interfacial PS films, and which particular drug features (size, hydrophobicity, charge) are associated with maximum efficiency of PS-assisted interfacial vehiculization.

In a final step, we tried to assess the contribution of exogenous surfactant reservoirs on the delivery of lipids and drugs over pre-existing endogenous surfactant films. It has been demonstrated that the spreading of exogenous surfactant is reduced by the presence of the endogenous film [44,45]. This vehiculization balance setup presents the limitation that no compression–expansion cycles can be introduced at the recipient compartment, which have been demonstrated as essential to promote material exclusion from the interface, continuously creating space for exogenous supply to replace it [12,46]. Still, we wanted to evaluate whether the spreading of surface-associated reservoirs could transport enough material to the recipient site to be detected, even under static conditions. The increment in surface pressure observed in both compartments upon the introduction of new material (Figure 6) indicates that the material adsorbed pushes the existing surfactant film, increasing its lateral compression, but not causing enough new material to reach the recipient trough. These results were also compared with experiments with surfactant-pre-coated interfaces connected by a paper bridge, to avoid the spreading of 3D structures (not shown). However, almost no fluorescence was detected in any of the experiments performed, regardless of the presence or absence of the bridge. By no means do these results indicate that surface-associated reservoirs are not playing an important role in surfactant spreading or increasing the transference of material over interfaces pre-coated by surfactant, but the static conditions used in this study impose a limitation to getting enough material delivered at the recipient compartment. A more complex design of this setup is under production to introduce the essential dynamism required to fully mimic the scenario in vivo. Cycling experiments in fully interconnected interfaces will allow us to evaluate whether surfactant reservoirs spread laterally following a compression–exclusion gradient contributing to transport large amounts of material.

In summary, the data presented here demonstrate that surface-associated structures formed upon surfactant adsorption are transported over the air–liquid interface during spreading. These essential surfactant components could play a critical role in the interfacial drug delivery mediated by PS, as they could be responsible for massive drug transference over the respiratory surface in inhaled therapies. Our results also point out the importance of including the hydrophobic surfactant proteins, or peptide fragments having good enough surface activity, into PS formulations designed for drug delivery, in order to ensure good spreading performance and the formation of the surface-associated structures that potentiate the delivery [5,47]. The double-surface balance introduced here will allow us to perform further experiments in which mucins, serum, or other relevant molecules in different physiological contexts could be incorporated into the path to study the behaviour of surfactant films and their associated structures and spreading rates in different relevant pathophysiological situations. As mentioned before, the introduction of a device to compress and expand the recipient interface would expand the possibilities and would facilitate the study of the interfacial therapy mediated by PS in vitro under the most physiological conditions. Some additional limitations should also be taken into account for future studies. The architecture of the respiratory system, temperature, time scale, mode of injection, and the concentration and volume of the samples applied in these experiments do not emulate how a drug/PS formulation would be administered in vivo and travel through the lungs. Therefore, these aspects should be further explored and considered to fully assess the relevance of surface-associated surfactant-assisted drug delivery under the most physiologically relevant scenario.

## Figures and Tables

**Figure 1 pharmaceutics-15-00256-f001:**
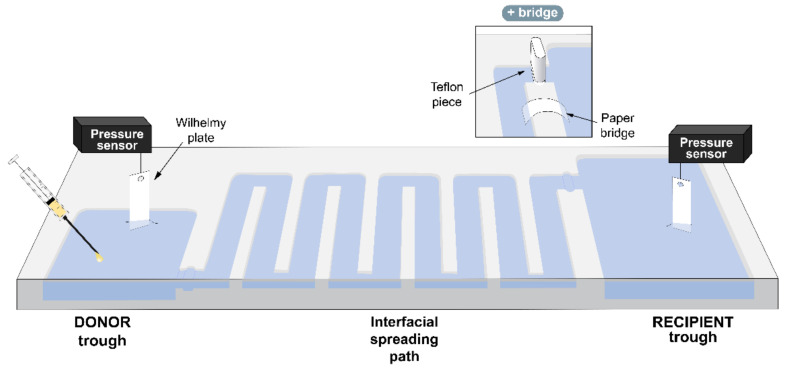
Schematic representation of the continuous double-balance setup. The vehiculization surface balance was designed and built on a stainless-steel plate. A donor compartment (24 cm^2^) is connected to a recipient compartment (45 cm^2^) by a long interfacial spreading path (90 cm, 75 cm^2^). The material is injected into the donor trough, and its adsorption and spreading are monitored as it produces changes in surface pressure by two surface pressure sensors. Both troughs and the path can be isolated one from each other by Teflon pieces designed to be introduced into gaps dug at the entrance of the compartments. When necessary, the access to the recipient compartment can be closed by the Teflon piece and a hydrated paper bridge can be used instead to connect this trough with the path (represented in the upper square). The figure shows a top view of the balance, as well as a cross section to illustrate the depth of the different compartments.

**Figure 2 pharmaceutics-15-00256-f002:**
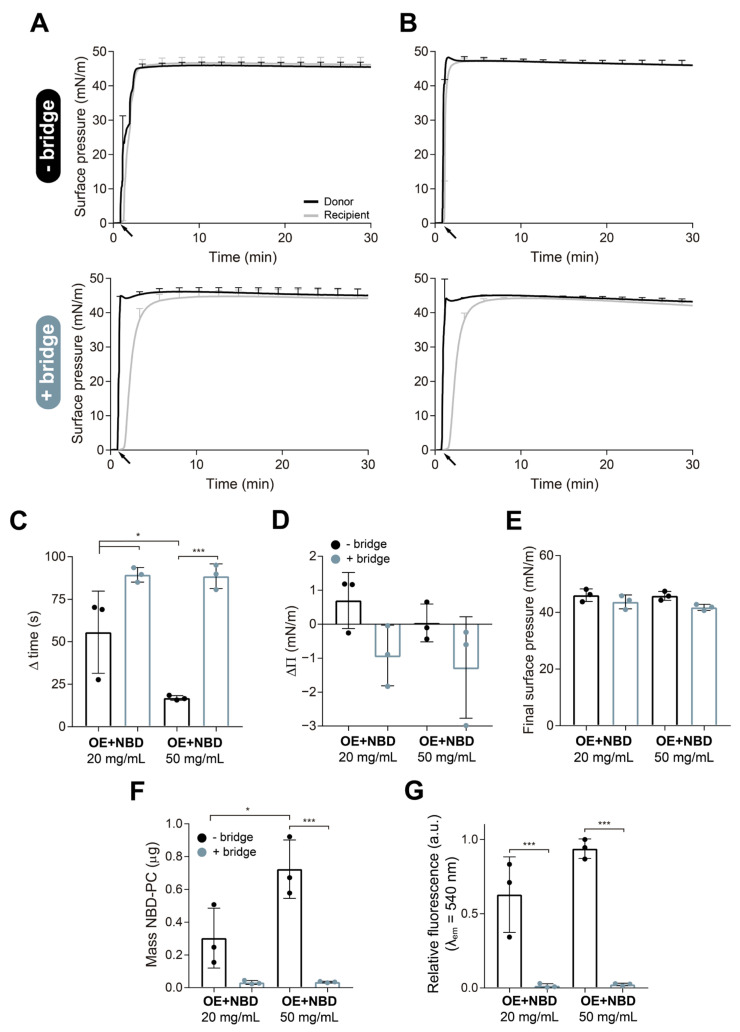
Analysis of the effect of surfactant concentration on interfacial spreading. Adsorption/spreading isotherms derived from the interfacial injection of 15 μL of an aqueous suspension of organic extract from native surfactant (OE) labelled with 1% (mol/mol) NBD-PC, at (**A**) 20 mg/mL or (**B**) 50 mg/mL, at the donor compartment of the vehiculization double-balance design. Experiments at both concentrations were performed in the absence (upper panels) and in the presence (bottom panels) of the paper bridge. Surface pressure was monitored for 30 min at the donor (black line) and recipient (grey line) compartments. Black arrows indicate material injection onto the donor interface. (**C**) Delay in the increase in surface pressure at the recipient compartment, measured as the time difference in seconds between the injection of the material onto the donor interface and the pressure increase in the recipient compartment until it reaches the half maximal pressure (Π_50_). The delay has been compared as obtained in experiments with (blue bars) and without (black bars) the bridge, and at both OE concentrations. (**D**) Difference in surface pressure between recipient and donor compartments (ΔΠ) registered 30 min after sample injection. Negative results mean that the recipient surface pressure was lower than measured at the donor compartment. (**E**) Final surface pressures reached at the recipient compartment 30 min after material addition at the donor trough. (**F**) Mass of NBD-PC (µg) collected from the recipient interface calculated upon interpolation of the fluorescence measured at λ_em_ = 540 nm into standard curves of known concentrations of NBD-PC. (**G**) Relative fluorescence emission at λ_em_ = 540 nm of the NBD-PC collected from the recipient subphase, after interfacial film collection, at the end of each experiment. Data represent mean and standard deviation calculated from a minimum of three independent experiments. One-way ANOVA followed by Tukey’s post-hoc test: (**C**) *p* < 0.001; (**F**) *p* < 0.001; (**G**) *p* < 0.001. (*) *p* < 0.05 and > 0.01, and (***) *p* ≤ 0.005.

**Figure 3 pharmaceutics-15-00256-f003:**
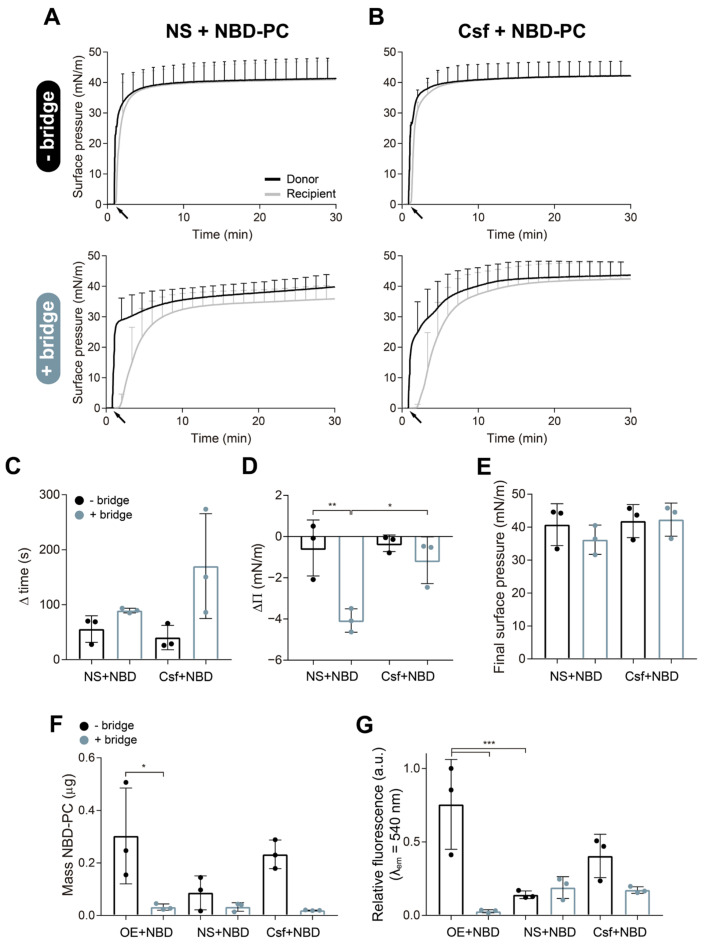
Effect of the interfacial bridge on the spreading and membrane transference of different materials over the air–liquid interface. Surface pressure (Π)-time isotherms upon interfacial injection of 15 μL of (**A**) native surfactant (NS) or (**B**) Curosurf (Csf) labelled with 1% (mol/mol) NBD-PC, at 20 mg/mL, at the donor compartment, performed in the absence (upper panels) and in the presence (bottom panels) of the paper bridge. Black arrows indicate material injection onto the donor interface. (**C**) Delay in the increase of surface pressure at the recipient compartment until reaching Π_50_ with the two materials tested, either with (blue bars) or without (black bars) the bridge. (**D**) Difference in surface pressure between recipient and donor compartments (ΔΠ) registered 30 min after injection of NS or Csf. (**E**) Final surface pressures reached at the recipient compartment 30 min after material addition at the donor trough. (**F**) Mass of NBD-PC (µg) collected from the recipient interface by aspiration calculated from a standard curve. Data from the aqueous suspension of OE have been included in this graph for comparison with the other material. (**G**) Relative fluorescence emission of NBD-PC at λ_em_ = 540 nm measured at the recipient subphase. Fluorescence from experiments with aqueous suspension of OE has been included in this graph for comparison. Data represent mean and standard deviation calculated from a minimum of three independent experiments. One-way ANOVA followed by Tukey’s post-hoc test: (**D**) *p* = 0.004; (**F**) *p* = 0.004; (**G**) *p* = 0.001. (*) *p* < 0.05 and > 0.01, (**) *p* ≤ 0.01 and > 0.005, and (***) *p* ≤ 0.005.

**Figure 4 pharmaceutics-15-00256-f004:**
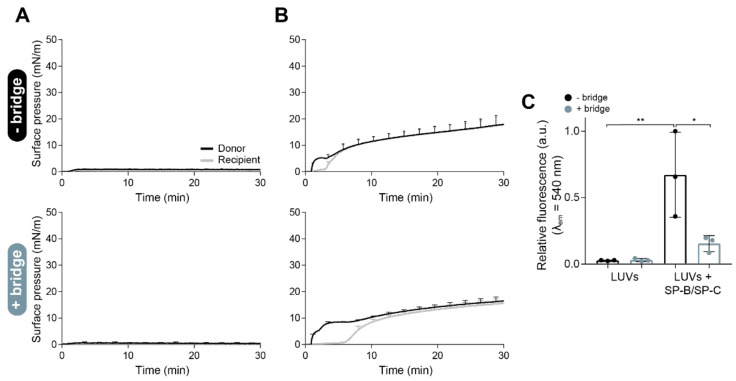
Interfacial spreading of lipidic or proteo-lipidic liposomes over the double-balance setup. Π-time isotherms of large unilamellar vesicles (LUVs) made of (**A**) DPPC:POPG (7:3, *w*/*w*) and (**B**) DPPC:POPG (7:3, *w*/*w*) + 2% (*w*/*w*) SP-B/SP-C (15 µL, 20 mg/mL) labelled with 1% (mol/mol) NBD-PC, both without (upper graphs) and with (bottom graphs) the paper bridge connecting the recipient trough with the rest of the system. (**C**) Relative fluorescence emission of the NBD-PC collected by aspiration from the recipient interface at the end of each experiment. Mean and standard deviation from three independent replicas are represented. One-way ANOVA (*p* = 0.004) followed by Tukey’s post-hoc test: (*) *p* < 0.05 and > 0.01, and (**) *p* ≤ 0.01 and > 0.005.

**Figure 5 pharmaceutics-15-00256-f005:**
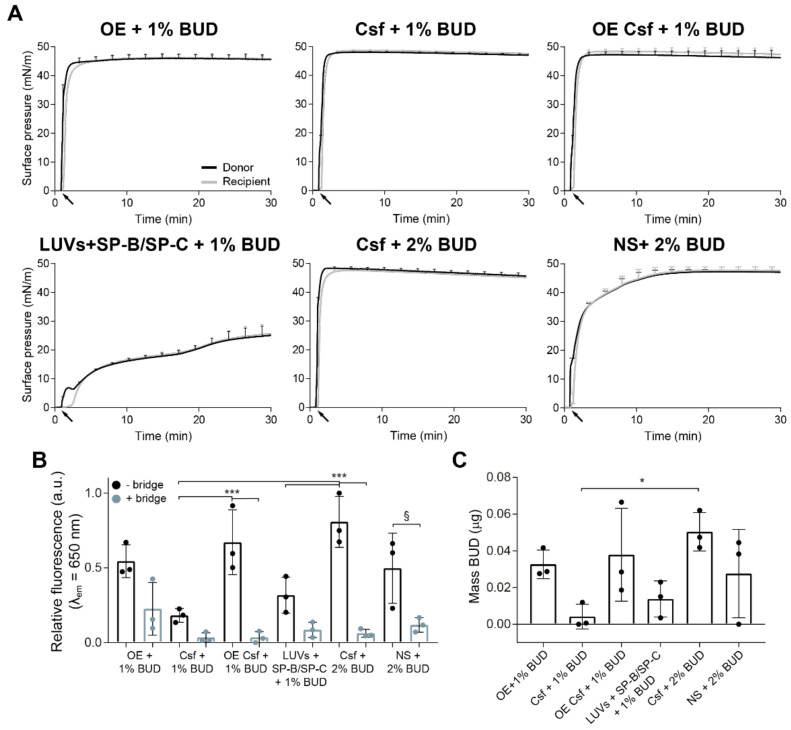
Interfacial vehiculization of budesonide (BUD) through pulmonary surfactant preparations. (**A**) Adsorption/spreading isotherms of aqueous suspensions of OE (top left), Csf (top middle), aqueous suspension of an OE of Csf (top right) and proteo-lipid LUVs (bottom left) combined with 1% (*w*/*w* with respect to total mass of phospholipids) BUD, and Csf (bottom middle), and NS (bottom right) in combination with 2% (*w*/*w*) BUD. Fifteen μL of each material at 20 mg/mL were deposited dropwise onto the donor interface (black arrows) and the changes in surface pressure were monitored during 30 min in both compartments. (**B**) Relative fluorescence emission of F-BUD (BUD) at λ_em_ = 650 nm at the recipient interfacial film collected by aspiration at the end of each experiment. Comparison of the fluorescence obtained for the different materials in experiments without (black bars) and with (blue bars) the bridge. (**C**) Comparison of the amount in µg of BUD transported by different materials to the recipient interface in experiments performed without the paper bridge. Data represents mean and standard deviation from at least three replicas. One-way ANOVA followed by Tukey’s post-hoc test: (**B**) *p* < 0.001; (**C**) *p* = 0.043. (§) *p* = 0.05, (*) *p* < 0.05 and > 0.01 and (***) *p* ≤ 0.005.

**Figure 6 pharmaceutics-15-00256-f006:**
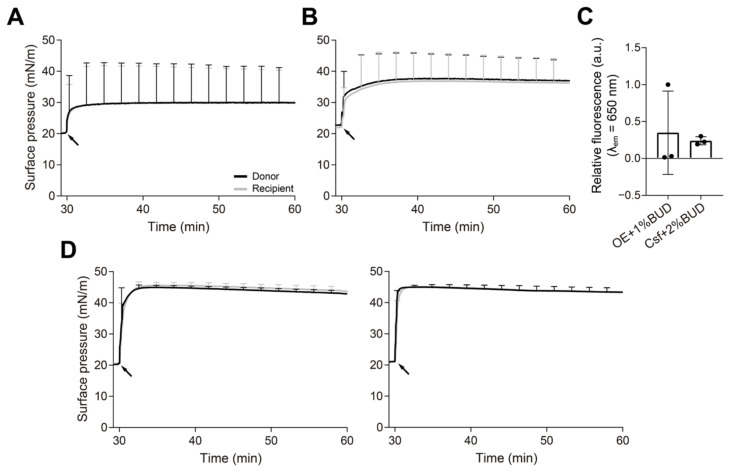
Interfacial spreading of PS over surfactant-occupied interfaces. Π-time isotherms of aqueous suspensions of (**A**) OE + 1% (*w*/*w*) BUD and (**B**) Curosurf + 2% (*w*/*w*) BUD, injected at time 30 min into the donor subphase (black arrows) below a pre-existing film of OE (injected as an organic solution to avoid formation of surface-associated reservoirs). (**C**) Relative fluorescence emission of F-BUD (BUD, λ_em_ = 650 nm) vehiculized by OE or Csf to the recipient interface. (**D**) Adsorption/spreading isotherms of 15 µL of OE + 1% (mol/mol) NBD-PC at 20 mg/mL (left graph) or 50 mg/mL (right graph) injected into the donor subphase of a pre-coated interface at 20 mN/m with OE in organic solvent. Black arrows indicate the material addition 30 min after covering the interface. Data show average and standard deviation of three independent experiments.

**Figure 7 pharmaceutics-15-00256-f007:**
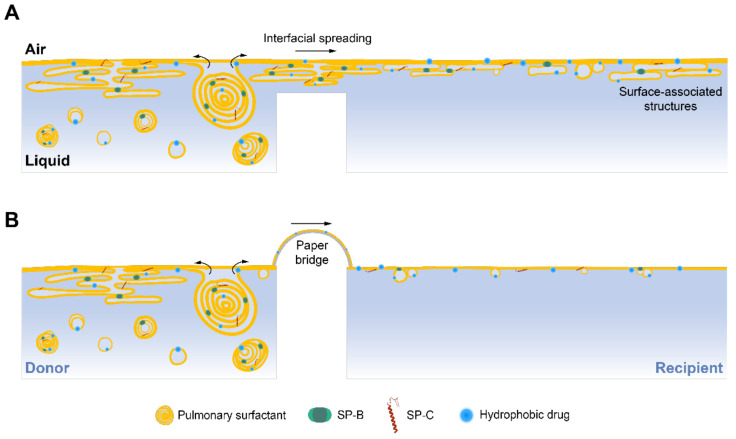
Proposed model of the interfacial spreading of surfactant structures in the double-balance setup. (**A**) Adsorption of a concentrated aqueous suspension of surfactant into the donor air–liquid interface forms surface-associated reservoirs, which can spread over the interface to the recipient compartment in the absence of the paper bridge. The spreading of surface-associated vesicles and multi-layered structures together with the interfacial film transport high amounts of lipids and the drug incorporated into them. (**B**) The structures associated with the surface formed upon surfactant adsorption are retained in the donor compartment in the presence of the bridge. Only the interfacial film and few associated vesicles could cross the bridge and reach the recipient interface, transporting less material. The drawings do not intend to represent the real scale of the system nor the structures.

## Data Availability

Not applicable.

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
