# Peer review of "Beyond the Interface: Improved Pulmonary Surfactant-Assisted Drug Delivery through Surface-Associated Structures"

_pharmaceutics, 2023, doi:10.3390/pharmaceutics15010256_

Round 1

Reviewer 1 Report

This manuscript by dr Garcia-Mouton and colleagues describes an experimental comparison between two methods used to study surfactant driven drug delivery. Using a newly designed through, with a long channel between a donor and recipient compartments they explore the effect of a paper bridge in deliver surfactant to the recipient side of the trough. Experiments are performed under a variety of conditions leading to the conclusion that the paper bridge interferes with drug delivery.  The study appears to be well performed, and the results are interesting.  However, I have concerns about several aspects of the paper.

Comments:

Title:  The title sounds fancy but really doesn’t say anything about the actual study. A better descriptive title is warranted  

Abstract: the abstract is very unbalanced with too much rationalization of the study and conclusions, but no information about results.

Line 22:  I fail to see how the new balance represents a 3D rather than a 2D structure.

Methods:

 Line 107-`08 sreplaces “lavages with “washes”

Line 120: during should be for

Line 141: reword “somehow emulating”, it makes it sound like magic.

Whey was a buffer without calcium utilized, considering that many functions of surfactant are enhanced by calcium?

Why were the experiments performed at room temperature?

Results:

Statistical notification on the graphs needs to be improved. The location of the * varies and connected brackets are confusing.  In addition, statistical significance should only be indicated for relevant comparisons and not those differing in two variables. For example, in figure 1C comparing values of the 20mg with bridge data to 50mg/ml without bridge is not a valid (or useful) comparison.  

It would be useful to present the data as scatter plots instead of bar graphs.

The authors should refrain from too much discussion in the result section. For example, line 248-252  as well as many other places, the authors provide an interpretation of the data that belongs in the discussion.

It would be useful to present data on the fluorescence in all compartments, rather than only in the recipient compartment, this will provide a more complete picture. In addition, it would be very valuable to have fluorescence data on multiple time-points rather than only after 30 min. It appears that, in many conditions, equilibrium surface pressure is obtained after a few minutes, it would be interesting to note if this also indicates complete transfer of fluorescence, specifically for budesonide.

Discussion:

Overall, the data supports the concept that from a methological point of view, a paper bridge results in less transfer of surfactant and budesonide to the recipient compartment, as compared to no bridge. The authors, should be a bit more careful and not over interpret the data.  For example, the idea that the paper bridge liquid film is too thin for surface associated structures (see figure 7) seems far-fetched and not based on data. My understanding of such bridge would be that the liquid film would be much ticker than the nm-size structures that are associated with the surfactant film.  

The author should address the limitations of there model. Temperature, time frame of experiments relative to in vivo, etc.

Author Response

TO REVIEWER 1

This manuscript by dr Garcia-Mouton and colleagues describes an experimental comparison between two methods used to study surfactant driven drug delivery. Using a newly designed through, with a long channel between a donor and recipient compartments they explore the effect of a paper bridge in deliver surfactant to the recipient side of the trough. Experiments are performed under a variety of conditions leading to the conclusion that the paper bridge interferes with drug delivery.  The study appears to be well performed, and the results are interesting.  However, I have concerns about several aspects of the paper.

Response: We sincerely thank the reviewer for his/her positive assessment of the manuscript.

Comments:

Comment 1: Title:  The title sounds fancy but really doesn’t say anything about the actual study. A better descriptive title is warranted.

Response: According to this suggestion, we have changed the title to “Beyond the interface: improved pulmonary surfactant-assisted drug vehiculization through surface-associated structures”.

Comment 2: Abstract: the abstract is very unbalanced with too much rationalization of the study and conclusions, but no information about results.

Response: The abstract has been modified accordingly.

Comment 3: Line 22:  I fail to see how the new balance represents a 3D rather than a 2D structure.

Response: We thank the reviewer for this comment. As briefly commented in the abstract and explained in the introduction, it has been demonstrated that the paper bridge only allows a very simple layer of surfactant to cross it, therefore, acting as a resistance against the lateral spreading of 3D-associated membranous structures (protruding towards the subphase, the “third dimension” of the surface film). Our novel design presented here connects donor and recipient compartments by a thick layer of water, permitting a true 3D connection (extended to surface-associated protruded structures) and not only a strict 2D layer like the one moving through the paper bridge. We have explained this a bit better in the abstract.

Methods:

Comment 4: Line 107-`08 replaces “lavages with “washes”

Response: The text has been changed as suggested.

Comment 5: Line 120: during should be for

Response: Corrected.

Comment 6: Line 141: reword “somehow emulating”, it makes it sound like magic.

Response: This has been changed to “emulating in a simplified way”.

Comment 7: Whey was a buffer without calcium utilized, considering that many functions of surfactant are enhanced by calcium?

Response: The presence of calcium in the alveolar spaces is important for surfactant activities, as the reviewer accurately indicates, including the function of surfactant hydrophilic proteins SP-A and SP-D, and other processes such as the secretion of surfactant by ATII cells or the conversion of secreted surfactant into tubular myelin figures. However, the presence of calcium ions in the buffer of the experiments in vitro originates additional effects in the behaviour of lipid suspensions that affect in different complex ways the structure of surfactant films as formed in vitro. In order to simplify our model and get clear outputs comparing the different conditions tested, the buffer used does not contains calcium. It remains to be investigated how much other factors such as calcium, temperature, or the presence of other elements such as the hydrophilic surfactant proteins modulate the interfacial spreading of surfactant structures. We have explicitly stated now the limitation and future perspectives to complete the study in the Discussion.

Comment 8: Why were the experiments performed at room temperature?

Response: We thank the reviewer for this relevant question. Although it could be more accurate in principle for translation into in vivo context, performing experiments with surface balances like the ones used here at 37C would entails some complications. The subphase could be thermostated at 37C, but the actual temperature at the air-liquid interface would be cooler as a consequence of evaporation. Evaporation of the subphase also affects surface pressure measurements and originates some problems to maintain stable the liquid connection through the path. For all these reasons and to simplify the model to a maximum, we have run all the experiments here at 25C. We are persuaded that the differences in spreading between 25C and 37C do not affect the final conclusion of the work. However, as stated in the answer to Comment 7, the future extension of this study should also include the analysis of the effect of temperature on the lateral spreading of surfactant films and their associated structures.

Results:

Comment 9: Statistical notification on the graphs needs to be improved. The location of the * varies and connected brackets are confusing.  In addition, statistical significance should only be indicated for relevant comparisons and not those differing in two variables. For example, in figure 1C comparing values of the 20mg with bridge data to 50mg/ml without bridge is not a valid (or useful) comparison. 

Response: We agree with the reviewer that this mode of notification was confusing, and some comparisons were not correct, so these issues have been properly addressed in all the figures.

Comment 10: It would be useful to present the data as scatter plots instead of bar graphs.

Response: Graphs have been modified accordingly, representing the data as scatter dots with bars, representing mean and SD.

Comment 11: The authors should refrain from too much discussion in the result section. For example, line 248-252  as well as many other places, the authors provide an interpretation of the data that belongs in the discussion.

Response: We completely agree with this comment. As suggested, we have removed these type of statements from the results, limiting them strictly to the discussion.

Comment 12: It would be useful to present data on the fluorescence in all compartments, rather than only in the recipient compartment, this will provide a more complete picture. In addition, it would be very valuable to have fluorescence data on multiple time-points rather than only after 30 min. It appears that, in many conditions, equilibrium surface pressure is obtained after a few minutes, it would be interesting to note if this also indicates complete transfer of fluorescence, specifically for budesonide.

Response: We thank the reviewer for these suggestions. The fluorescence data from the other compartments have not been measured because it has been done in previous studies and no differences were observed between different conditions, even though there were significant differences in the recipient compartment. As the material is injected in excess, the donor interface is fully saturated in the different formulations.

Multiple time-points measurement would give very valuable information and will be considered for future studies. The problem of the current approach is that each time-point has to be obtained from a different experiment, because fluoresce measurements are taken from the whole material extracted from the recipient compartment, and would considerably increase the number of experiment in the study. Collection at 30 min ensures enough time in the different conditions to transport measurable amounts of material into the recipient compartment, while also ensuring that the passive diffusion is not affecting the results.

Discussion:

Comment 13: Overall, the data supports the concept that from a methological point of view, a paper bridge results in less transfer of surfactant and budesonide to the recipient compartment, as compared to no bridge. The authors, should be a bit more careful and not over interpret the data.  For example, the idea that the paper bridge liquid film is too thin for surface associated structures (see figure 7) seems far-fetched and not based on data. My understanding of such bridge would be that the liquid film would be much ticker than the nm-size structures that are associated with the surfactant film.  

Response: We have modified some sentences regarding the role of the bridge that may be confusing (lines 529-534). As commented by the reviewer, it is very unlikely that the hydration layer is thinner than the 3D reservoirs. However, it has been demonstrated by Yu and Possmayer that only a surfactant monolayer crosses this bridge, and this is supported by the results presented in this manuscript. As surfactant travels through the air-liquid interface and should not be affected by a net formed by the cellulose, one hypothesis could be that the hydration layer is thin enough to retain some dense surfactant reservoirs and only led some vesicles and simpler structures to cross it. However, this effect should be further studied as discussed in the manuscript. Figure 7 is an illustrative representation of the system and the effect produced by the bridge, but it does not represent the real scale of the structures (this has been now explicitly indicated in the figure caption).

Comment 14: The author should address the limitations of there model. Temperature, time frame of experiments relative to in vivo, etc.

Response: Some of the limitations of our system were already included in the manuscript. However, we have included some lines in the discussion stating some additional limitations of our study.

Reviewer 2 Report

The authors propose a new experimental design using two Langmuir donor/recipient troughs connected by a long channel to study pulmonary surfactant-mediated transport of budesonide through the interfacial layer and the surfactant-associated three-dimensional reservoirs. They compare measurements made with their system with those obtained with the wet bridge device they (and others) have used in previous studies. Then they use their new double trough to study budesonide transport.

Although the new system and the final objective are very interesting, I find the article too confusing to be published as it stands. Basically, it contains two studies that could both be further investigated and published separately. Many experiments have been carried out, but it is often difficult to understand what is being learned from them. There are too many different systems and conditions. The message should be simplified. It is not clear why it is necessary to compare conditions with and without a wet bridge, especially as these experiments provide little convincing information about the transfer, and the authors do not show the results of experiments with budesonide carried out with the wet bridge.

As far as form is concerned, I think that it would be better to bring together the results and the discussion, as many of the experiments described in the Results are justified by the Discussion of other experiments. All graphs within a figure should be identified by a letter or a number. This would facilitate understanding of the legend and text, especially when different systems are studied, as in Figure 5. In Figure 4 A/B, I suggest using the same Y-axis scale as in Figures 2 and 3 to allow a better comparison of the results.

1-What is known about the thickness of the associated structures of the interfacial film, and the thickness of the liquid layer that passes over the wet bridge? And what about the interaction of the various film components with the wet paper? The authors mention the possible role of the ultrathin hydration layer in line 527, and the effect of paper material, in line 530, but they don't comment on them.

2-In Figure 2D, why is the surface pressure of the recipient "even higher" than that of the donor? What is the accuracy of the measurements made by the surface pressure sensors?  I did not see this information in the experimental section.

3-The fact that the authors only work with surfactant concentrations leading to maximum lowering of the surface tension makes it very difficult to know what does and does not pass the wet bridge. Although a temporal shift in the pressure isotherm is observed in Fig. 2A/2B, the final surface pressure hardly changes; 1 mN/m is not significant, as the authors acknowledge (lines 224-226). NBD-PC can mix with the surfactant, but it cannot model the whole surfactant. The text in lines 227-251 and the discussion of figure 3 do not seem to me to prove much about what is transferred and what is not. In my opinion, the only information that can be derived from the experiments depicted in Figures 2 and 3 is that whatever surfactant is studied, the wet bridge causes a delay in the spreading of the surfactant in the recipient trough.

4- I don't understand what the authors mean in lines 334-335: "a mere passive diffusion from the donor to the recipient trough along the volume of the whole subphase instead of caused by an efficient interface-assisted spreading of the interfacial film and the associated structures".

5- The associated structures are dynamic systems and in the presence of a new interface, such as that offered by the second trough, they could be drawn towards the interface and integrated into the expanding interfacial film. In such conditions, there would be no associated structures/multilayer when the film reaches the second trough. There would be only a monolayer. Let's say that the associated structures are maintained because there is enough surfactant to saturate all the interface. As they are integral parts of the surfactant layer, I expect the film to be destabilized if they do not pass through the wet bridge. This is apparently not the case with the simplest "surfactants" such as OE for which only the delay is seen, probably because there is a large excess of surfactant. On the other hand, the effect observed with NS and Csf is very interesting (Fig 3) but little commented on by the authors: it seems to show a destabilization that is compensated for Curosurf, but not for NS, probably because NS contains water-soluble protein fraction, unlike Curosurf. Could the authors comment on these results?

6- The authors chose liposomes in the subphase as a model for the surfactant-associated structures. In Figure 4, the difference in NBD-PC concentration between the system with a bridge and the system with no bridge is not clear. Why would NBD-PC be in the subphase? Because the vesicles do not adsorb and barely interact with the surfactant layer? Liposomes seem to me hardly comparable with the multilayers formed by the surfactant when the layer is condensed. They are not reservoirs in the sense of “surfactant reservoirs”. Conversely, the multilayers are an integral part of the interfacial film as shown in Figure 7, even if the components are not in the plane of the interface. I do not dispute that the wet bridge does not allow these structures to pass whereas the set-up proposed by the authors does. I think that the experiments carried out here do not allow this to be demonstrated unambiguously.

7-  I find it difficult to understand the results without little information on the surface properties of budesonide and its interaction with lipids and surfactant proteins. How surface active is budesonide? How does it interact with the lipid and peptide fractions?

8-How was the path length chosen? From the results, it seems that the time for the surface pressure gradient effect is too short for rearrangements to take place at the interface, especially in a spontaneous way without exerting lateral pressure with a barrier.

9-Line 440: I do not understand the sentence: "This result could indicate that the surfactant reservoirs did not spread over the pre-existing film". If the authors are studying the penetration of the surfactant molecules into the spread film, why would there be spreading over?

10-In their comment on Figure 6D, the authors indicate that there was no fluorescence measured in the recipient trough. Did they check the fluorescence of NBD-PC in the donor compartment?

11- I don't understand the sentence on lines 482-485: where did the authors see "massive transference of surfactant and a surplus of therapeutic molecules"?

12- Line 516: Increasing OE concentration from 20 mg/mL to 50 mg/mL did not produce great differences in surface pressure values (Figure 2)": I do not see how it could be otherwise if the 20 mg/mL deposit is already saturating the interface. In Figure 6, when the deposited layer is not saturating the interface, the surfactant in the subphase can penetrate into the film and the surface pressure increases.

13-Line 547: The fluorescence results for NBD are not very convincing especially as NBD is added to liposomes that are merely dispersed in the subphase. It is difficult to draw a clear and consistent conclusion from these results.

14- Line 568: BUD is better carried by OE: To what do the authors attribute this? Greater interaction with lipids? 

Author Response

To REVIEWER 2

The authors propose a new experimental design using two Langmuir donor/recipient troughs connected by a long channel to study pulmonary surfactant-mediated transport of budesonide through the interfacial layer and the surfactant-associated three-dimensional reservoirs. They compare measurements made with their system with those obtained with the wet bridge device they (and others) have used in previous studies. Then they use their new double trough to study budesonide transport.

Although the new system and the final objective are very interesting, I find the article too confusing to be published as it stands. Basically, it contains two studies that could both be further investigated and published separately. Many experiments have been carried out, but it is often difficult to understand what is being learned from them. There are too many different systems and conditions. The message should be simplified. It is not clear why it is necessary to compare conditions with and without a wet bridge, especially as these experiments provide little convincing information about the transfer, and the authors do not show the results of experiments with budesonide carried out with the wet bridge.

As far as form is concerned, I think that it would be better to bring together the results and the discussion, as many of the experiments described in the Results are justified by the Discussion of other experiments. All graphs within a figure should be identified by a letter or a number. This would facilitate understanding of the legend and text, especially when different systems are studied, as in Figure 5. In Figure 4 A/B, I suggest using the same Y-axis scale as in Figures 2 and 3 to allow a better comparison of the results.

Response: We thank the reviewer for his/her detailed revision and the nice comments and suggestions. The manuscript aims to evaluate the spreading of surface-associated surfactant reservoirs over the air-liquid interface and its contribution to the delivery of drug, which, to the best of our knowledge, has not been studied before. We think that complementing and comparing the results about the transport of lipids and the delivery of drugs together in the same manuscript facilitates the understanding of the message, including the comprehension of the mechanisms by which this interface-assisted drug delivery occurs, and increases the scope of the work. Moreover, as this manuscript also presents the new device developed, the different experiments also illustrate its potential and limitations.

Comparing experiments with and without the bridge was essential to demonstrate that previous designs using double-surface balances were actually underestimating the capabilities of surfactant to transport lipids and drugs, as well as to proof the spreading of surface-associated 3D structures. The results with Budesonide carried out with the bridge are shown in Figure 5 and Figure S2 of the manuscript, and its delivery by PS in a double-balance setup using the paper bridge was already extensively evaluated by Hidalgo et al. (Langmuir, 2017).

We agree with the reviewer that the text could be made clearer, and for the sake of clarity, we have modified extensively Results and Discussion attending the reviewer suggestions. For instance, we have removed all the discussion from the Results section to make it clearer and easier to follow. All the graphs and panels have been identified with a letter, except for the isotherms that are identified directly with a label identifying the type of material used, as we think it is clearer and faster to recognize, without the need to look for the material in the figure caption. The Y-axis in the graphs have the same scale.

Comment 1: What is known about the thickness of the associated structures of the interfacial film, and the thickness of the liquid layer that passes over the wet bridge? And what about the interaction of the various film components with the wet paper? The authors mention the possible role of the ultrathin hydration layer in line 527, and the effect of paper material, in line 530, but they don't comment on them.

Response: Only few studies have addressed the structure of PS surface-associated reservoirs. The dynamic structure of PS during compression-expansion cycling makes difficult to evaluate the thickness of the surfactant layer, both considering the interfacial film exposed to air and the associated reservoirs. A lipid monolayer is around 2.5 nm thick, and the surfactant film including the reservoirs should be of few tens of nanometers. We do not know the thickness of the paper hydration layer, but we speculate that it might be thin enough to hinder the transfer of some dense reservoirs. As discussed in the manuscript, the cellulose matrix of the bridge can also interact in different ways, including by formation of hydrogen bonds with the hydrophilic regions of the lipids/surfactant, and this could contribute to retain some material, but the interaction of the lipids and other surfactant components with the cellulose matrix should be investigated to confirm this hypothesis. We have modified the text to state this matter.

Comment 2: In Figure 2D, why is the surface pressure of the recipient "even higher" than that of the donor? What is the accuracy of the measurements made by the surface pressure sensors?  I did not see this information in the experimental section.

Response: The surface pressure measured at the recipient trough when the bridge is not used was usually slightly higher than that of the donor probably as a consequence of a slight compression effect due to the ultrarapid spreading of material. However, the differences were of just 1-2 mN/m, close to the limit of significance. The accuracy of the sensor is 0.1 mN/m. This information has been included in the methods section of the revised manuscript.

Comment 3: The fact that the authors only work with surfactant concentrations leading to maximum lowering of the surface tension makes it very difficult to know what does and does not pass the wet bridge. Although a temporal shift in the pressure isotherm is observed in Fig. 2A/2B, the final surface pressure hardly changes; 1 mN/m is not significant, as the authors acknowledge (lines 224-226). NBD-PC can mix with the surfactant, but it cannot model the whole surfactant. The text in lines 227-251 and the discussion of figure 3 do not seem to me to prove much about what is transferred and what is not. In my opinion, the only information that can be derived from the experiments depicted in Figures 2 and 3 is that whatever surfactant is studied, the wet bridge causes a delay in the spreading of the surfactant in the recipient trough.

Response: We kindly thank the reviewer for these comments. The two concentrations tested were chosen to have an excess of surfactant and indeed saturate the interface, ensuring the formation of PS reservoirs. It also somehow mimics what is expected to occur locally at the point of surfactant delivery in cases of true surfactant therapy delivered into the upper airways. A lower concentration leading to limiting conditions could avoid or reduce the formation of the 3D structures whose study is a main goal here. Lower concentrations could also produce lower spreading rates and so the intrinsic differences in material transference (as seen comparing NBD and BUD fluorescence) could be originated in these spreading rates differences and not due to the presence or not of the reservoirs.

As the reviewer indicates, we agree that NBD-PC is by no mean modeling the whole surfactant. However, the aim of using this lipid dye is that it mixes well within surfactant membranes and serves as a marker to estimate how much surfactant travels from one compartment to the other. Also, NBD-PC labelling allows comparing the transfer of lipid-like molecules between different formulations. The experiments performed in Figures 2 and 3 aims to demonstrate that, apart from slowing down the spreading process and causing a delay, the bridge is reducing the mass of material transferred through it. The pressures reached by the different materials in the absence and presence of the bridge at time 30 min were very similar, so even though the spreading process was delayed, at time 30 min both conditions reached the same state “at the surface”. However, the mass of lipids and drug transported was significantly lower in most of the formulations. We have tried to explain better this interpretation into the Discussion.

Comment 4: I don't understand what the authors mean in lines 334-335: "a mere passive diffusion from the donor to the recipient trough along the volume of the whole subphase instead of caused by an efficient interface-assisted spreading of the interfacial film and the associated structures".

Response: We wanted to set the objective of these experiments. The use of LUVs without surfactant hydrophobic proteins was motivated by the fact that these vesicles do not adsorb at the air-liquid interface, and therefore, can only travel by diffusion through the liquid subphase. Therefore, the absence of fluorescence reaching the recipient trough in these experiments confirms that these vesicles are not traveling through the subphase (which does not depend on the interfacial activity) for the time of the experiments, and therefore, that our observations can only be due to the contribution of the adsorption and spreading of material associated to the air-liquid interface. We have explained this a bit more in the text.

Comment 5: The associated structures are dynamic systems and in the presence of a new interface, such as that offered by the second trough, they could be drawn towards the interface and integrated into the expanding interfacial film. In such conditions, there would be no associated structures/multilayer when the film reaches the second trough. There would be only a monolayer. Let's say that the associated structures are maintained because there is enough surfactant to saturate all the interface. As they are integral parts of the surfactant layer, I expect the film to be destabilized if they do not pass through the wet bridge. This is apparently not the case with the simplest "surfactants" such as OE for which only the delay is seen, probably because there is a large excess of surfactant. On the other hand, the effect observed with NS and Csf is very interesting (Fig 3) but little commented on by the authors: it seems to show a destabilization that is compensated for Curosurf, but not for NS, probably because NS contains water-soluble protein fraction, unlike Curosurf. Could the authors comment on these results?

Response: We agree with the reviewer, and it makes an interesting point of view. It is certainly possible that the retention effect produced by the bridge on the surfactant film could be associated to destabilization, which could be an additional contribution to the delay observed (this possibility has been now explicitly stated in the discussion of the revised manuscript, page 14). This delay is observed in the three formulations compared (OE, NS and Csf, observed in Fig 2 and 3), although with small differences between them, so the possible destabilization effect likely has an impact in all of them. The concentrations used were the same for the three samples, so the excess of surfactant should be similar. SP-A promotes PS adsorption and further connections between membranes, which could also contribute to the longer delay observed in NS or to a differential destabilization effect, as suggested by the reviewer. We have included additional discussion in this respect.

Comment 6: The authors chose liposomes in the subphase as a model for the surfactant-associated structures. In Figure 4, the difference in NBD-PC concentration between the system with a bridge and the system with no bridge is not clear. Why would NBD-PC be in the subphase? Because the vesicles do not adsorb and barely interact with the surfactant layer? Liposomes seem to me hardly comparable with the multilayers formed by the surfactant when the layer is condensed. They are not reservoirs in the sense of “surfactant reservoirs”. Conversely, the multilayers are an integral part of the interfacial film as shown in Figure 7, even if the components are not in the plane of the interface. I do not dispute that the wet bridge does not allow these structures to pass whereas the set-up proposed by the authors does. I think that the experiments carried out here do not allow this to be demonstrated unambiguously.

Response: The NBD-PC concentration upon delivery with pure lipidic liposomes (with no proteins) is almost 0 both with and without the bridge. As the reviewer indicates, there are no differences between these two conditions because the liposomes show no interfacial activity, and that is why there is no fluorescence in the interface or in the subphase (the fluorescence could only reach the subphase of the recipient through by mere passive diffusion through the liquid and not due to interfacial spreading).

The experiments with the liposomes were not intended to emulate the surfactant-reservoirs or the complex interfacial film formed by PS. Fully lipidic liposomes were selected as the negative control of no interfacial activity and no 3D structures. In the presence of the hydrophobic proteins, liposomes are able to adsorb and spread interfacially. These proteins are essential to promote formation of membrane-membrane contacts, which can lead to the formation of 3D surfactant-like reservoirs. These experiments were performed to demonstrate how simple formulations as the one used here can be used to deliver lipids/drugs, and how the bridge also affects simpler structures. We think our system demonstrates that the bridge (and, possibly, other structures that could be present in the in vivo context) avoid the transport of structures associated to the “more superficial” part of the surfactant film, which are supported by the connections created by SP-B and SP-C.

Comment 7:  I find it difficult to understand the results without little information on the surface properties of budesonide and its interaction with lipids and surfactant proteins. How surface active is budesonide? How does it interact with the lipid and peptide fractions?

Response: We have included a sentence in page 10 of the revised manuscript regarding the surface activity and effect of Budesonide over PS. As BUD is a hydrophobic corticosteroid, it interacts with the acyl chains of the lipids and the hydrophobic moieties of the proteins present in surfactant membranes. Its incorporation into membranes and PS has been extensively studied before.

Comment 8: How was the path length chosen? From the results, it seems that the time for the surface pressure gradient effect is too short for rearrangements to take place at the interface, especially in a spontaneous way without exerting lateral pressure with a barrier.

Response: We thank the reviewer for the comment. The path and the design of the system does not certainly emulate the complex architecture of the respiratory system. This was designed to increase the length of the path in a minimal space. Increasing the length, together with a more complex shape could reduce even further the possibility of passive diffusion over the subphase. Moreover, longer path also helps to see larger differences between different conditions or samples. We may agree with the reviewer that transient changes in surface pressure during spreading may or may not be enough to cause effects on the interfacial structure of the materials, but in other setups we have been able to detect measurable pressure waves as a consequence of rapid spreading. All together points to the dynamic character of the surfactant-promoted spreading phenomena, which is surely further modulated in the in vivo scenario.

Comment 9: Line 440: I do not understand the sentence: "This result could indicate that the surfactant reservoirs did not spread over the pre-existing film". If the authors are studying the penetration of the surfactant molecules into the spread film, why would there be spreading over?

Response: The objective of these experiments was to determine whether the surfactant reservoirs or an extra addition of material from the subphase were able to insert into the pre-existing interfacial layer and traveling associated to the surface. As the surface pressure increases upon injection of this material, we hypothesized that the reservoirs could insert into the film and then spread over it until coating the whole surface with 3D structures. However, we could not confirm from our experiments whether this is actually happening or not. We assume that in the system used here, intrinsically static, the amount of fluorescent material that reaches the recipient trough over the possible “movement” of the reservoirs is not enough to be detected. We have added an statement to clarify this interpretation.

Comment 10: In their comment on Figure 6D, the authors indicate that there was no fluorescence measured in the recipient trough. Did they check the fluorescence of NBD-PC in the donor compartment?

Response: As explained also in comment 12, the fluorescence in the donor compartments was not measured in these experiments, but it was done in previous studies and no significant differences were detected between different conditions, even if there were differences in the recipient compartment measurements.

Comment 11: I don't understand the sentence on lines 482-485: where did the authors see "massive transference of surfactant and a surplus of therapeutic molecules"?

Response: This sentence has been changed to “expanding the transference of surfactant and a surplus of therapeutic molecules over long distances”.

Comment 12: Line 516: Increasing OE concentration from 20 mg/mL to 50 mg/mL did not produce great differences in surface pressure values (Figure 2)": I do not see how it could be otherwise if the 20 mg/mL deposit is already saturating the interface. In Figure 6, when the deposited layer is not saturating the interface, the surfactant in the subphase can penetrate into the film and the surface pressure increases.

Response: With this statement, we only wanted to make clear that the differences in fluorescence were not due to differences in the surface pressure. Which means that the interface is certainly saturated at both concentrations, leading indeed to similar surface pressure values; the differences in the transference of material should then necessarily be due to structures not affecting this pressure and thus associated to the interface.

Comment 13: Line 547: The fluorescence results for NBD are not very convincing especially as NBD is added to liposomes that are merely dispersed in the subphase. It is difficult to draw a clear and consistent conclusion from these results.

Response: As explained in comment 18, these experiments were performed as a negative control. NBD-PC was incorporated into the liposomes that, actually, did not adsorb into the air-liquid interface and are merely dispersed in the subphase. Through this, we wanted only to determine whether vesicles that are dispersed in the subphase could still be able to “travel” from the donor to the recipient trough, contributing to the delivery of material and, thus, whether the differences observed between absence vs. presence of the bridge could be due to this and not only to interfacial delivery.

Comment 14: Line 568: BUD is better carried by OE: To what do the authors attribute this? Greater interaction with lipids?

Response: As the different material tested are all formed by lipids (with similar composition, except for the vesicles that present simplified compositions in comparison with surfactant) we would not attribute this best vehiculization to the intrinsic interaction with (some) lipids. As commented in page 16 of the revised manuscript, the incorporation of a hydrophobic drug as BUD into an organic extraction preparation of surfactant is more efficient than in an pre-formed aqueous preparation of surfactant because both materials are dissolved in organic solvents, facilitating its mix and interaction.

Round 2

Reviewer 2 Report

Thanks to the authors for answering my questions. The article is clearer now.